# Economic evaluation of surgical treatments for women with stress urinary incontinence: a cost-utility and value of information analysis

Mehdi Javanbakht [1] ,[1] Eoin Moloney [1] ,[1] Miriam Brazzelli [1] ,[2] Sheila Wallace,[1] Laura Ternent,[1] Muhammad Imran Omar,[3] Ash Monga,[4] Lucky Saraswat,[5] Phil Mackie,[6] Frauke Becker,[7] Mari Imamura,[2] Jemma Hudson [1] ,[2] Michal Shimonovich,[2] Graeme MacLennan [1] ,[2] Luke Vale,[1] Dawn Craig[1]

For numbered affiliations see end of article.

**Correspondence to**
Eoin Moloney;
eoin.moloney@newcastle.ac.uk

## ABSTRACT

**Objectives** Stress urinary incontinence (SUI) and stress-predominant mixed urinary incontinence (MUI) are common conditions that can have a negative impact on the quality of life of patients and serious cost implications for healthcare providers. The objective of this study was to assess the cost-effectiveness of nine different surgical interventions for treatment of SUI and stress-predominant MUI from a National Health Service and personal social services perspective in the UK.

**Methods** A Markov microsimulation model was developed to compare the costs and effectiveness of nine surgical interventions. The model was informed by undertaking a systematic review of clinical effectiveness and network meta-analysis. The main clinical parameters in the model were the cure and incidence rates of complications after different interventions. The outcomes from the model were expressed in terms of cost per quality-adjusted life-years (QALYs) gained. In addition, expected value of perfect information (EVPI) analyses were conducted to quantify the main uncertainties facing decision-makers.

**Results** The base-case results suggest that retropubic mid-urethral sling (retro-MUS) is the most cost-effective surgical intervention over a 10-year and lifetime time horizon. The probabilistic results show that retro-MUS and traditional sling are the interventions with the highest probability of being cost-effective across all willingness-to-pay thresholds over a lifetime time horizon. The value of information analysis results suggest that the largest value appears to be in removing uncertainty around the incidence rates of complications, the relative treatment effectiveness and health utility values.

**Conclusions** Although retro-MUS appears, at this stage, to be a cost-effective intervention, research is needed on possible long-term complications of all surgical treatments to provide reassurance of safety, or earlier warning of unanticipated adverse effects. The value of information analysis supports the need, as a first step, for further research to improve our knowledge of the actual incidence of complications.

## INTRODUCTION

Stress urinary incontinence (SUI) in women is a disabling and common condition which

### Strengths and limitations of this study

► An economic model has been developed to consider the relative cost-effectiveness of nine different surgical interventions for stress urinary incontinence and stress-predominant mixed urinary incontinence in a UK setting.
► The model has been informed and populated based on a comprehensive systematic review and network meta-analysis of the clinical effectiveness of surgical interventions.
► Uncertainty in the model results has been explored through a probabilistic sensitivity analysis, and value of information analysis, to consider the value of future research.
► Due to limited long-term data on effectiveness, short-term data were extrapolated over a longer-term time horizon.

reduces quality of life and causes significant stress and concern among individual patients.[1][2] Commonly defined as the 'involuntary leakage of urine' due to activities such as coughing, sneezing or exercising, the prevalence of SUI varies (20% to 50%) but is greater in women who have had children and in older women.[3][4] Associated conditions include urge urinary incontinence (UUI), which is characterised by a sudden and uncontrollable need to urinate (ie, urgency), and mixed urinary incontinence (MUI) which refers to a combination of the symptoms associated with SUI and UUI.[2] Where the symptoms of SUI dominate those of UUI in a patient with MUI, stress-predominant MUI is said to exist.

In addition to the emotional impact that SUI can have on patients, there are serious cost implications associated with the condition, both for the healthcare provider and

the individual patient.[5] Among the previous studies to have investigated the economic burden of the condition, one published study based on data from three European countries suggested that the total cost of SUI is approximately £818 million in the UK,[2] while another suggested a healthcare cost to the UK National Health Service (NHS) of £117 million per year (SUI only).[5] In addition to the health service costs, patients may incur out-of-pocket expenses due to the need to purchase consumables and services not available through their healthcare provider.[5]

Conservative treatment with physiotherapy to deliver pelvic floor muscle exercises and bladder training is the first-line of treatment for patients with SUI. However, when this fails, or when insufficient relief from symptoms is achieved, surgery is the recommended treatment.[6] The aim of surgery is to support or partially obstruct the bladder neck and/or urethra, thus preventing the leakage of urine on exertion.[7] Surgical techniques have evolved over time, moving most recently to more minimally invasive techniques, such as mid-urethral slings (MUS) where a synthetic mesh or tape is placed under the urethra and secured using a number of different methods.[8] However, it is unclear if these newly available treatments such as retropubic mid-urethral sling (retro-MUS), single incision sling and injectable bulking agents really result in equivalent or better cost and health outcomes than older operations that were previously available (such as anterior vaginal repair or colposuspension).

The wide range of surgical operations available, the different techniques used to perform these operations and the lack of a consensus among surgeons regarding which approach to use, make it challenging to establish which procedure should be used to treat SUI or stress-predominant MUI. Health economic techniques have become increasingly important as a tool to help inform decisions between treatments and interventions. They allow for the consideration of all relevant costs (from the specified perspective) and clinical outcomes associated with the treatments being compared. Findings can then be used to make a decision on how best to allocate resources in the most cost-effective way. This study was conducted as part of a larger project to evaluate the cost-effectiveness of retro-MUS versus eight different comparator surgical interventions for the treatment of women with SUI or stress-predominant MUI.[9]

## METHODS
### Study design
A Markov microsimulation (MM) model was developed to assess cost-effectiveness. The model was informed by a review of published economic evaluations, and the findings of a systematic review of clinical effectiveness and network meta-analyses (including 120 trials) conducted as part of the larger project to estimate the relative effectiveness of nine different surgical treatments.[9] Retro-MUS was chosen as the intervention against which all other surgical interventions would be compared, due to the fact

that it is the most common type of surgery and it is widely regarded as standard practice.[10] The eight comparators were: (1) anterior vaginal repair or anterior colporrhaphy (anterior repair), (2) bladder neck needle suspensions (bladder neck needle), (3) open abdominal retropubic colposuspension (open-colpo), (4) laparoscopic retropubic colposuspension (lap-colpo), (5) traditional suburethral retropubic sling procedures (trad-sling), (6) transobturator mid-urethral sling (transob-MUS), (7) single incision sling procedures (single incision sling) and (8) periurethral injection bulking agents (injectable agents).

Health outcomes from the model were expressed in terms of quality-adjusted life-years (QALYs) and costs in 2018/2019 GBP (£). The costs were estimated from the NHS and personal social services perspective. Both costs and QALYs were evaluated over 1-year, 10-year and lifetime time horizons and discounted using a 3.5% annual discount rate.[11] The expected cost and QALYs for each of the strategies were compared using incremental cost-effectiveness ratios (ICERs) where appropriate.

### Target population
The model was based on a hypothetical group of women (45 to 55 years) with SUI or stress-predominant MUI, undergoing surgical intervention for their condition.

### Model structure
The model structure is presented in figure 1. Treatment history for both SUI/stress-predominant MUI and UUI was recorded for each modelled woman and used to define the patient transitions to different health states and treatment types. The model assumes that patients can receive a maximum of three surgical treatments for treatment of SUI/stress-predominant MUI, which includes the initial surgery plus two subsequent re-treatments. MUS can be offered after the failure of all surgery types, and therefore re-treatment was always assumed to be either retro-MUS (55%) or transob-MUS (45%). If all three surgeries fail, patients have to manage their symptoms using containment products. The model allows for individuals to elect to use containment products at any point after the initial treatment has failed. Patients with stress-predominant MUI who still have UUI after SUI is cured, or those who develop UUI due to a surgery, that is, as a complication following the procedure, will receive three lines of treatment based on clinical guidelines including first-line (bladder training), second-line (medication) and third-line treatment (botulinum toxin A). The model was developed in TreeAge Pro package (TreeAge Software, Inc, Williamstown, Massachusetts, USA).[12]

### Model inputs
#### Effectiveness of surgical treatments
The relative effectiveness of surgical treatments, in terms of subjective cure rates, were based on the results of a network meta-analysis.[9] Table 1 describes the mean and median ORs for different surgical treatments versus

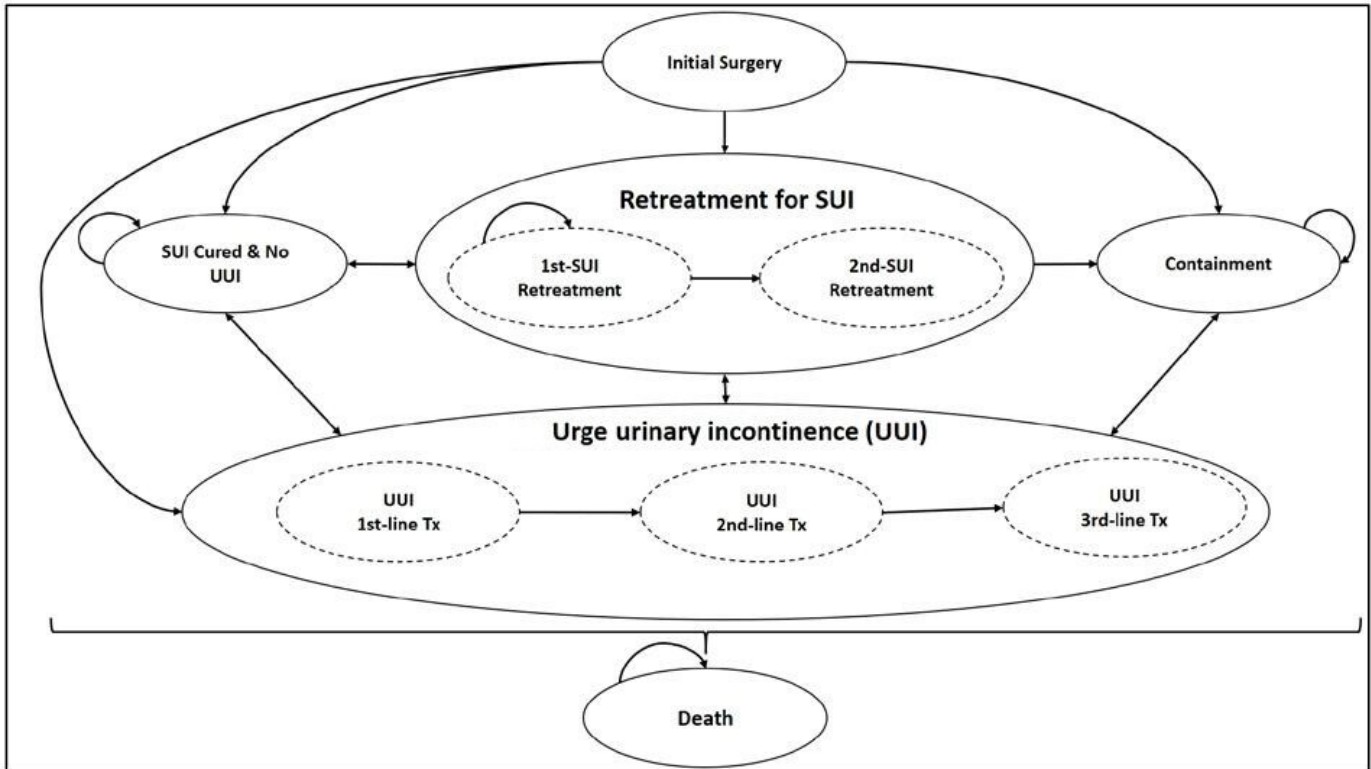

**Figure 1** Markov model structure. SUI, stress urinary incontinence.

retro-MUS. The absolute cure rates were calculated in the model by combining the information on relative cure rates described in Table 1 with the absolute cure rates for retro-MUS, presented in table 2. The long-term recurrence rates after retro-MUS were estimated using parametric survival models. Scale and shape parameters were estimated for a Weibull hazard function using the reported cure rates at 1 year and 5 years (online supplementary table S1, online supplementary figure S1, online supplementary table S2 and online supplementary figure S2, respectively). In the base-case analysis, it was assumed that 75% of women whose first treatment was not successful would seek re-treatment and 30% of women

whose first re-treatment failed would seek a second re-treatment.[13] Due to data limitations, it was not possible to estimate the success rates of re-treatment following failure of prior surgery. Therefore, in the model it was assumed that subsequent surgeries (retro-MUS or transob-MUS) were 90% as effective as when administered as a primary surgery.

### Complication and mortality rates after surgical treatments

The model incorporated only severe complications/ adverse events that are most important in terms of their effect on patient's quality of life, cost and duration of impact. To estimate complication incidence rates,

| Table 1 | ORs of cure rates for surgical interventions compared with retro-MUS | | | |
|---|---|---|---|---|
| Treatment | Mean | Median | Lower bound of 95% CrI | Upper bound of 95% CrI |
| Transob-MUS versus retro-MUS | 0.742 | 0.738 | 0.588 | 0.923 |
| Open-colpo versus retro-MUS | 0.874 | 0.853 | 0.544 | 1.325 |
| Lap-colpo versus retro-MUS | 0.605 | 0.58 | 0.315 | 1.046 |
| Trad-sling versus retro-MUS | 1.106 | 1.061 | 0.623 | 1.846 |
| Single incision versus retro-MUS | 0.511 | 0.504 | 0.36 | 0.699 |
| Bladder neck needle versus retro-MUS | 0.368 | 0.34 | 0.154 | 0.745 |
| Anterior repair versus retro-MUS | 0.235 | 0.22 | 0.105 | 0.452 |

anterior repair, anterior vaginal repair (anterior colporrhaphy); bladder neck needle, bladder neck needle suspensions; CrI, credible interval; lap-colpo, laparoscopic retropubic colposuspension; MUS, mid-urethral sling; open-colpo, open abdominal retropubic colposuspension; retro-MUS, retropubic mid-urethral sling; single incision, single incision sling procedures ('mini-slings'); trad-sling, traditional suburethral retropubic sling procedures; transob-MUS, transobturator mid-urethral sling.

**Table 2** Estimation of absolute cure rates after retro-MUS in different time points - results from meta-analysis

| Time | Median | 95% CrI | Number of studies | Number of participants |
|------|--------|---------|-------------------|------------------------|
| 6 months | 0.776 | (0.175 to 0.983) | 17 | 908 |
| 12 months | 0.841 | (0.214 to 0.990) | 44 | 2882 |
| 24 months | 0.784 | (0.454 to 0.941) | 6 | 315 |
| 36 months | 0.341 | (0.001 to 0.995) | 5 | 205 |
| 60 months | 0.329 | (0.005 to 0.979) | 3 | 377 |

CrI, credible interval; retro-MUS, retropubic mid-urethral sling.

random-effects meta-analysis models were fitted to data on complication rates from the randomised controlled trials identified within the wider systematic review.[14] All of the complications included in the model, and estimated incidence rates, are presented in online supplementary table S3. In the model, these complication rates were defined as distributions (beta distribution). Where evidence for particular adverse events associated with specific procedures were not identified, it was assumed that this complication would not occur. Age-specific all-cause mortality rates were derived from general population mortality statistics reported in national life tables.[15] The effectiveness of the three lines of treatment for UUI were informed by the results from previous meta-analyses.[6 16]

### Resource use and unit costs
Surgeries for SUI vary in terms of the complexity of the procedure and the setting in which surgery is conducted. For patients undergoing anterior repair, bladder neck needle, open-colpo, lap-colpo and trad-sling, surgery would typically be conducted in an inpatient setting. Patients undergoing retro-MUS, transob-MUS, single incision sling and injectable agents would typically be treated in a day-case setting. In addition to the cost of individual surgeries, costs associated with complementary tests, treatments and consultations that would typically be carried out in advance of, and following, each surgery were also considered. Unit costs were derived from UK NHS reference costs,[17] Personal Social Services Research Unit[18] and British National Formulary[19] for medication. The price year of the analysis was 2018/2019. Where sources used to derive costs were from prior to 2019, these costs were inflated to a 2019 price year. All costs are reported in online supplementary table S4.

### Health utility
The baseline health utility value for pre-treatment SUI was derived from a previous UK-based economic evaluation comparing two types of surgical intervention for this condition.[20] The utility value for a successful treatment was derived from a UK-based study exploring health outcomes in women with urinary incontinence.[21] Utility decrements associated with each complication included

in the model were obtained from previous studies[22 23] (online supplementary table S5).

### Patient and public involvement
As part of the larger project,[9] patient and public involvement (PPI) was included to understand patient's thoughts and feelings regarding their condition and treatment and to help add context to the review and meta-analysis of clinical effectiveness. However, no additional PPI was sought for this modelling work.

### Analysis and sensitivity analyses
Both deterministic and probabilistic sensitivity analyses (PSA) were used to explore parameter and other forms of uncertainty surrounding the estimates of cost-effectiveness. Deterministic sensitivity analyses including; applying higher incidence rates of mesh complications, longer durations of persistent pain complication, higher incidence rates for persistent pain complication post MUS procedures and applying different values for short-term and long-term cure rates after retropubic MUS were performed to determine the impact of changing key parameters and assumptions on the model results. The PSA was run with 1000 simulations for each patient, and cost-effectiveness acceptability curves (CEACs) were produced in order to identify the probability of the different surgeries being cost-effective across a range of willingness-to-pay (WTP) thresholds. The economic model was used to quantify the main uncertainties facing decision-makers and to help inform decisions about the direction of future research. This was explored through value of information (VOI) analysis methods: expected value of perfect information (EVPI) and expected value of partial perfect information (EVPPI) analysis. In total, it was assumed that in the UK 15 000 surgical treatments are conducted annually for the treatment of SUI.

## RESULTS
### Base-case analysis
The base-case analysis results are presented in table 3, and in the form of a CEAC (lifetime time horizon) in figure 2. The table reports strategies from the least to the most costly. Over a lifetime time horizon, retro-MUS is, on average, the least costly (£8666) and the second most effective (24.005 QALYs) surgical treatment. Trad-sling is more costly (+ £405) but also more effective (+0.009) than retro-MUS over the lifetime; however, it is not cost-effective, based on the National Institute for Health and Care Excellence WTP threshold, with an ICER of £45 340. All other surgical treatments are dominated, as they are more costly and less effective than retro-MUS. Both retro-MUS and trad-sling have similarly high probabilities of being cost-effective at £20 000 (51% and 43%, respectively) and £30 000 (48% and 45%, respectively) WTP thresholds over a lifetime time horizon. Over a 10-year time horizon, retro-MUS is the dominant strategy with a greater than 90% probability of being cost-effective

**Table 3** Results of the probabilistic analysis at 1 year, 10 years and lifetime time horizons

| Time horizon | Strategy | Cost (£) | Incremental cost (£) | QALY | Incremental QALY | ICER (£) (Δcost/ΔQALY) | Probability of being cost-effective at different threshold (%) | |
|---|---|---|---|---|---|---|---|---|
| | | | | | | | £20 000 | £30 000 |
| 1 year | Single incision sling | 1844 | | 0.764 | | | 100% | 100% |
| | Transob-MUS | 2470 | 626 | 0.752 | –0.012 | Dominated | 0% | 0% |
| | Retro-MUS | 2490 | 646 | 0.752 | –0.012 | Dominated | 0% | 0% |
| | Bladder neck needle | 2682 | 838 | 0.757 | –0.006 | Dominated | 0% | 0% |
| | Injectable agents | 2705 | 861 | 0.742 | –0.021 | Dominated | 0% | 0% |
| | Trad-sling | 2941 | 1097 | 0.725 | –0.038 | Dominated | 0% | 0% |
| | Anterior repair | 2955 | 1111 | 0.766 | 0.002 | 522 756 | 0% | 0% |
| | Open-colpo | 4847 | 1891 | 0.775 | 0.009 | 212 116 | 0% | 0% |
| | Lap-colpo | 4875 | 28 | 0.765 | –0.010 | Dominated | 0% | 0% |
| 10 years | Retro-MUS | 4905 | | 7.270 | | | 95.0% | 93.0% |
| | Single incision sling | 5271 | 366 | 7.023 | –0.248 | Dominated | 0% | 0% |
| | Trad-sling | 5471 | 566 | 7.215 | –0.056 | Dominated | 5.0% | 7.0% |
| | Transob-MUS | 5519 | 614 | 7.107 | –0.163 | Dominated | 0% | 0% |
| | Injectable agents | 5882 | 977 | 7.068 | –0.202 | Dominated | 0% | 0% |
| | Bladder neck needle | 6101 | 1197 | 7.025 | –0.245 | Dominated | 0% | 0% |
| | Anterior repair | 6629 | 1725 | 6.967 | –0.303 | Dominated | 0% | 0% |
| | Open-colpo | 7596 | 2691 | 7.204 | –0.066 | Dominated | 0% | 0% |
| | Lap-colpo | 7997 | 3092 | 7.106 | –0.164 | Dominated | 0% | 0% |
| Lifetime | Retro-MUS | 8666 | | 24.005 | | | 51.0% | 48.0% |
| | Trad-sling | 9071 | 405 | 24.014 | 0.009 | 45 340 | 43.0% | 45.0% |
| | Transob-MUS | 10 174 | 1103 | 23.435 | –0.580 | Dominated | 0% | 0% |
| | Single incision sling | 10 189 | 1118 | 23.221 | –0.793 | Dominated | 0% | 0% |
| | Injectable agents | 10 292 | 1221 | 23.512 | –0.503 | Dominated | 0% | 0% |
| | Bladder neck needle | 10 803 | 1732 | 23.312 | –0.702 | Dominated | 0% | 0% |
| | Open-colpo | 11 605 | 2535 | 23.839 | –0.175 | Dominated | 6.0% | 7.0% |
| | Anterior repair | 11 609 | 2539 | 23.168 | –0.847 | Dominated | 0% | 0% |
| | Lap-colpo | 12 440 | 3369 | 23.522 | –0.492 | Dominated | 0% | 0% |

anterior repair, anterior vaginal repair (anterior colporrhaphy); bladder neck needle, bladder neck needle suspensions; ICER, incremental cost-effectiveness ratio; lap-colpo, laparoscopic retropubic colposuspension; open-colpo, open abdominal retropubic colposuspension; QALY, quality-adjusted life-year; retro-MUS, retropubic mid-urethral sling; single incision sling, single incision sling procedures ('mini-slings'); trad-sling, traditional suburethral retropubic sling procedures; transob-MUS, transobturator mid-urethral sling.

at the £20 000 and £30 000 WTP thresholds presented. However, over a 1-year time horizon single incision sling is dominant compared with all other strategies, except for anterior repair and open-colpo, which are both marginally more effective.

Figure 2 shows that over a lifetime time horizon, both trad-sling and retro-MUS have a high probability of being cost-effective at all WTP threshold values presented. Given the number of comparators, if the interventions were comparable we would expect an 11% chance of each being cost-effective. The only other strategy with a reasonably sized probability of being cost-effective over the lifetime is open-colpo (6% and 7% probabilities of being cost-effective at £20 000 and £30 000 WTP

thresholds, respectively). No other strategy has a significant probability of being cost-effective at any of the WTP thresholds presented.

### Sensitivity analysis and VOI analysis
In the first sensitivity analysis (SA), results from the study by Keltie et al[24] were used to inform the mesh complications incidence rates after retro-MUS and transob-MUS. Results from this SA show that retro-MUS remains the most cost-effective option. However, when the incidence rate of mesh complications following retro-MUS and transob-MUS is increased to 10%, and then 20%, trad-sling becomes the most cost-effective intervention (online supplementary table S6). Results from the second SA

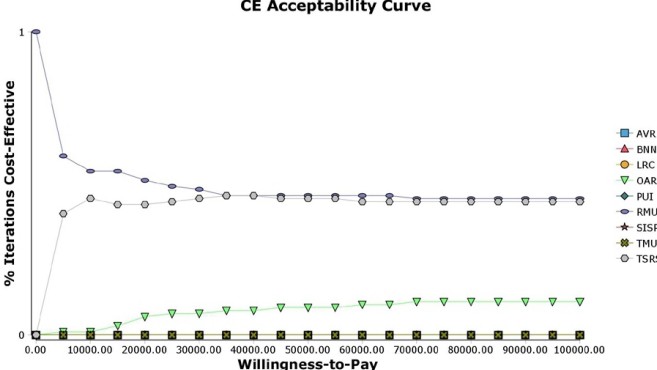

**Figure 2** Cost-effectiveness acceptability curves for the nine surgical treatments: lifetime time horizon. AVR, anterior vaginal repair; BNNS, bladder neck needle suspension; CE, cost-effectiveness; LRC, laparoscopic retropubic colposuspension; OARC, open abdominal retropubic colposuspension; PUI, periurethral injection; RMUS, retropubic mid-urethral sling; SISP, single incision sling procedure; TMUS, transobturator mid-urethral sling; TSRS, traditional sub-urethral retropubic sling.

show that when a longer duration for persistent pain is assumed (3 and 5 years), the probability of retro-MUS being the most cost-effective option decreases and the respective probabilities for trad-sling and open-colpo increase. Furthermore, results from the third SA show that when a higher incidence rate (10% and 20%) of persistent pain after retro-MUS and transob-MUS is assumed, the probability of retro-MUS being the most cost-effective option decreases and the respective probability for trad-sling increases. Detailed results from all sensitivity analyses are reported in online supplementary table S6–S10, respectively.

The VOI analyses results show that the EVPI per patient is £11 857 and the estimated EVPI for the UK population for 1 year is £177.9 million. This figure increases as the time horizon (or period of time over which the information would be useful) is increased (table 4). Results from

**Table 4** Results from expected value of information analysis

| The expected value of removing all current decision uncertainty | Overall EVPI (£) | Overall EVPI (QALY) |
| --- | --- | --- |
| Per person affected by the decision | 11 857 | 0.56 |
| Per year in UK assuming 15 000 persons affected per year | 177 855 000 | 8385 |
| Over 5 years | 889 275 000 | 41 930 |
| Over 10 years | 1 778 550 000 | 83 850 |
| Over 15 years | 2 667 825 000 | 125 800 |
| Over 20 years | 3 557 100 000 | 167 700 |

EVPI, expected value of perfect information; QALY, quality-adjusted life-year.

the EVPPI analysis also show that the largest value appears to be in removing uncertainty around the incidence rate of complications and treatment effectiveness parameters. Results from this analysis are presented in online supplementary table S11.

## DISCUSSION
To our knowledge, this economic evaluation is the most comprehensive assessment of the cost-effectiveness of surgical interventions for the treatment of SUI or stress-predominant MUI in women. While the broader study on which this paper is based[9] presents the entire spectrum of evidence available on the clinical effectiveness, safety and cost-effectiveness of surgical treatments for SUI, the work presented here focusses specifically on the economic viability of these treatments. Additionally, given that the cost estimates included in the previous study are now outdated, we have considered the relative cost-effectiveness of alternative interventions using cost information from the most recent price year (2019). Due to limited long-term clinical data, the results are uncertain but suggest that retro-MUS is the least costly and most cost-effective surgical intervention over 10-year and lifetime time horizons; results which are largely driven by the low initial cost of retro-MUS. This is primarily due to the fact that this procedure is conducted in a day-case setting, and there is a lower chance of needing repeat surgery due to its higher cure rate compared with all other surgical treatments (except for trad-sling). Trad-sling is the one alternative intervention, which has a relatively high probability of being cost-effective over 10-year and lifetime time horizons. This is likely driven by the high success rate associated with trad-sling, relative to retro-MUS. Over a 1-year time horizon, retro-MUS is dominated (more costly and less effective) by single incision sling, primarily due to the low initial cost of this procedure and the low complication rate associated with this type of surgery.

One of the strengths of our economic model is the evaluation of *nine* different surgical treatments in one study, informed by data from a comprehensive evidence synthesis and network meta-analysis where all the direct and indirect evidence (120 trials) were used to estimate the relative effectiveness of different surgical treatments in terms of cure rates.[9] Very few cost-effectiveness studies in this clinical area have included complications within their analysis, despite the fact that the incidence of each complication can have an impact on the woman's quality of life, and result in costs incurred for the health system. Therefore, in the present study the impact that complications such as infection, urge urinary incontinence, voiding difficulties, bladder or urethral perforation, mesh removal, short-term pain and persistent pain have on costs and effects have been incorporated into the model and explored in sensitivity analysis. Approximately 50% of women who have SUI[4] also suffer from UUI symptoms; given that UUI potentially affects the woman's quality of life more than SUI,[25] treatment pathways associated with

UUI are also incorporated in the model to more accurately estimate QALYs for this patient group. The analysis was conducted in accordance with the International Society of Pharmacoeconomics and Outcomes Research modelling guidelines[26] and extensive sensitivity analyses were undertaken to explore and characterise uncertainty in the model parameters. Our results were generally robust to the sensitivity analyses performed and are comparable to the findings of other published studies.

There are a number of published cost-effectiveness analyses evaluating some of the surgeries that we have assessed, the general findings of which are presented to allow for comparison. Two studies compared the single incision versus mid-urethral sling procedures over a 1-year time horizon, concluding that single incision was less costly and of similar effectiveness.[27 28] A UK study compared the cost-effectiveness of retro-MUS (tension-free vaginal tape (TVT)) versus open colposuspension, laparoscopic colposuspension, trad-sling and injectable agents, concluding that TVT dominated open-colpo over a 5-year time horizon.[29] A cost-utility analysis in the UK to assess the cost-effectiveness of TVT compared with open Burch colposuspension found that TVT was less costly and more effective.[20] A further study to assess the cost-effectiveness of TVT versus laparoscopic mesh colposuspension concluded that TVT was more cost-effective over a 1-year time horizon.[30] Although there are limitations with many of these studies and heterogeneity present in the methods, the results from all of the above studies are largely in agreement with the findings from our economic model; generally supporting the conclusion that retro-MUS (TVT) is likely to be the most cost-effective option when compared with the other types of surgeries for the treatment of SUI and stress-predominant MUI.

Results from the VOI analysis indicate that further research should focus on adverse events that, however rare, can have a detrimental impact on women's quality of life when they do occur (eg, tape extrusion/exposure).

### Limitations
One of the main limitations of the current study is the lack of long-term data which necessitated the extrapolation of relatively short-term data to 10 years, and over the lifetime of women included in the analysis. Therefore, the results presented would only apply in a situation where relative differences in the effectiveness of retro-MUS compared with the comparators do not change with longer follow-up. The long-term incidence of complications after the surgical treatments are also currently unknown. All of the estimated incidence rates for complications after each surgical treatment were based on data from trials with relatively short-term follow-up times. We tested the impact of possible higher incidence rates of some of the complications on the results, to incorporate and explore some of this uncertainty within our model. Finally, the economic analysis included costs to the NHS only. As highlighted earlier, individual patients also buy different products and incur the costs of containment products themselves; a future analysis may also need to consider these out-of-pocket costs to patients.

### CONCLUSIONS
The results of this study suggest that, in the medium-term and long-term, retro-MUS is less costly and more effective than all other surgical interventions for the treatment of SUI; therefore, it is a dominant strategy. However, the results should be interpreted with caution as the long-term performance of all surgical treatments in terms of both continence and unanticipated adverse effects is not reliably known. Therefore, the results presented are based on the extrapolation of short-term and medium-term evidence.

**Author affiliations**
[1]Population Health Sciences Institute, Newcastle University, Newcastle upon Tyne, UK
[2]Health Services Research Unit, University of Aberdeen, Aberdeen, UK
[3]Academic Urology Unit/Cochrane Incontinence Group, University of Aberdeen, Aberdeen, UK
[4]Gynaecology, University Hospitals Southampton Foundation Trust, Southampton, UK
[5]Obstetrics and Gynaecology, Aberdeen Royal Infirmary, Aberdeen, UK
[6]Scottish Public Health Network, NHS Health Scotland, Glasgow, UK
[7]Health Economics Research Centre, University of Oxford, Oxford, UK

**Acknowledgements** We would like to acknowledge all those involved in the wider study exploring the effectiveness and cost-effectiveness of surgical treatments for women with stress urinary incontinence.

**Contributors** MJ, DC, EM, LT and LV designed the study. MJ and EM built the model and ran the analyses. DC, MB, LS, PM, JH, GM and LV helped to inform the model. All co-authors (EM, MB, SW, LT, MIO, AM, LS, PM, FB, MI, JH, MS, GM, LV, DC) commented on the draft version of the manuscript and approved its final version.

**Funding** This research was commissioned by the NIHR HTA Programme as project number 15/09/06. The views expressed are those of the authors and not necessarily those of the NHS, the NIHR or the Department of Health and Social Care, UK. The funders were not actively involved in the research process at any stage. The study design, collection, analysis and interpretation of data, the writing of the manuscript and the decision to submit it for publication were all performed independent of the funders.

**Competing interests** None declared.

**Patient consent for publication** Not required.

**Provenance and peer review** Not commissioned; externally peer reviewed.

**Data availability statement** All data relevant to the study are included in the article or uploaded as supplementary information.

**ORCID iDs**
Mehdi Javanbakht http://orcid.org/0000-0002-8661-8439
Eoin Moloney http://orcid.org/0000-0002-3025-5413
Miriam Brazzelli http://orcid.org/0000-0002-7576-6751
Jemma Hudson http://orcid.org/0000-0002-6440-6419
Graeme MacLennan http://orcid.org/0000-0002-1039-5646

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
