## [Reviewer comments · BMJ Open]

ARTICLE DETAILS

TITLE (PROVISIONAL)	Economic Evaluation of Surgical Treatments for Women with Stress Urinary Incontinence: A Cost-Utility and Value of Information Analysis
AUTHORS	Javanbakht, Mehdi; Moloney, Eoin; Brazzelli, Miriam; Wallace, Sheila; Ternent, Laura; Omar, Muhammad; Monga, Ash; Saraswat, L; Mackie, Phil; Becker, Frauke; Imamura, Mari; Hudson, Jemma; Shimonovich, Michal; MacLennan, Graeme; Vale, Luke; Craig, Dawn

VERSION 1 – REVIEW

REVIEWER	S.E. Zwolsman Netherlands
REVIEW RETURNED	02-Dec-2019

GENERAL COMMENTS	Some references are outdated.
-------------------------------

REVIEWER	Sue Ross University of Alberta, Canada
REVIEW RETURNED	03-Dec-2019

GENERAL COMMENTS	Please complete reference # 9 Thanks for inviting me to review this very interesting paper by Javanbakht and colleagues. It provides a fascinating overview of the information available on surgical interventions for stress urinary incontinence (SUI), a condition that adversely affects many women. Surgery for SUI in women has been a rapidly evolving clinical field, dominated by the development of new and improved devices in a highly competitive surgical device market. This setting produced many new devices and procedures which have been marketed with little evidence of safety or effectiveness. Some devices were apparently removed from the market because of concerns about safety, and some have been/are the subject of expensive class action lawsuits. The harmful outcomes of surgery have been a source of concern for women and surgeons, attracting much media attention. In this environment where the relative costs of treatments are coming under increasing scrutiny, the paper under review provides a carefully designed cost-effectiveness analysis using a Markov microsimulation model incorporating the results from a systematic review of clinical effectiveness and network meta-analyses. The model is based on available evidence about possible relative costs
---

	and outcomes of nine different surgical interventions. The paper adds cost-utility and value of information analyses. The authors suggest, based on their research, that retropubic mid-urethral surgery is a cost-effective intervention compared to other surgical approaches, but point out that further research is needed. The intended audience of this paper is unclear, but it will certainly be of interest to other health economists and health services researchers. Limitations The paper is limited only by the quantity, quality and age of the evidence available. In particular, the long-term outcomes for SUI surgery are unknown and therefore it is impossible to estimate the outcomes accurately. This is acknowledged by the authors, and they have carried out sophisticated modelling and analyses to address uncertainty as well as they can. The surgical management of SUI has been changing rapidly over the past two decades and continues to do so. Over that time, the changes were driven largely by the surgical device industry. More recently, concerns of women (heightened by adverse media attention) have changed the public (and political) view of SUI surgery. The ongoing development of new devices is likely to continue, perhaps leading to improved outcomes over time. On the other hand, the move away from mesh surgery has been led by women. This has led to a recent increase in fascial sling surgery which requires a hospital stay, unlike mesh surgery such as retropubic mesh sling surgery which can be done as a day procedure. Extrapolating the economic evidence from older research to current date is fraught with uncertainty because of the wide variety of available choices that can influence the type and cost of pelvic floor surgery carried out (including surgical technique or approach, materials, devices, hospital/clinic length of stay). As with all such economic analyses, the concern of the reader is about the completeness, accuracy and validity of the original information available to incorporate into economic models. It is often an unattainable goal to achieve good data on long-term outcomes of surgery, because individual devices or interventions are withdrawn or abandoned after a short life cycle, particularly in a manufacturer-led device market (Health Policy and Technology (2015) 4, 168-188). The authors have addressed the fact that they are unable to fully predict the long-term outcomes of the surgery. Strengths The research team is ideally placed to carry out this research. The team includes highly respected health economists, experts in guideline development and systematic reviews, and clinicians. The authors carried out the ESTER research which forms the base of the current research. Thus this multidisciplinary group incorporates the essential knowledge and expertise to undertake the research. The research itself is well designed and conducted, and it provides additional analysis based on a large systematic review of published literature and economic evaluation of surgical treatments for SUI (ESTER). The inclusion of nine commonly used management options is novel, useful and thought-provoking. The paper is clearly written.
--	---

	The authors are very careful not to over-interpret the value of their findings. They caution that more research is needed, specifically with respect to incidence and impact of long-term complications. The design of the research by Javanbakht et al, offers some hope that long-term outcomes might be available, because it groups together types of procedure (rather than select specific devices). Even so, the long-term evaluation of outcomes of "new" surgery will inevitably be out of date before the research reaches its long-term outcomes, with many new devices introduced (and withdrawn) over the years. Modelling studies such as this are the best way to at least predict what might happen in the future. As well, they are able to compare the differences in cost-utility between particular approaches by applying standardised approaches. A significant strength of the paper is in describing the methods used to design and undertake the research. This provides a practical example of how such an economic evaluation could be done in another clinical area. As well, the ESTER HTA report describes in further detail the methods used. Suggestion for the authors I have just one suggestion for the authors. I agree that this economic evaluation is probably "the most comprehensive assessment of the cost effectiveness of surgical interventions for the treatment of SUI or stress-predominant MUI in women", however this conclusion is similar to that in the HTA report. A brief description about how this current paper [Javanbakht et al] complements and enhances the scope of the HTA report [Brazzelli et al], plus the recent BMJ paper [Imamura et al], would help the reader understand why this particular paper is important and relevant. This comment is likely best in the introduction or discussion where you could explain clearly how your findings are unique within the context of the ESTER work, rather than apparently duplicating some of what has been done. It is a huge strength of this paper that it forms part of such a significant body of work.
--	--

REVIEWER	Edna Keeney University of Bristol, UK
REVIEW RETURNED	20-Jan-2020

GENERAL COMMENTS	The costs used are from 2015/16 GBP. This is a minor point but it should be relatively easy to update these to the latest price year. It is not immediately clear from the text and tables how the success rates for each of the re-treatments were derived. The cure rates for the initial treatment are presented as odds ratios, are these assumed to be the same for subsequent surgeries? Related to this, it would be good to have a table of the absolute cure rates for each of the surgical interventions as these are more easily interpreted by the reader than odds ratios. A rationale should be given as to why Retro-MUS is not the most cost-effective over a 1 year time horizon but is over the 10 year and lifetime time horizons.
---

	It could be made clearer in the results and the discussion that there seems to be very little difference between Retro-MUS and Trad-sling. The CEAC would suggest that, if anything, Trad-sling appears to be the most cost-effective at a £20-£30k threshold. This might be confusing for the reader if not discussed adequately. The QALYs in Table 3 should be given to three decimal places. Why are the results from the EVPPI analysis not presented in Table 4? How were the absolute complications incidence rates derived? Was Retro-MUS again used as the baseline intervention? What did you do about the lack of complications data for some interventions? There are formatting issues with Table S6, S8 and S9 as some headings have been cut off.
--	---

VERSION 1 – AUTHOR RESPONSE

Response from authors to Reviewer 1:

(1) We have not referenced any work prior to year 2000, with the majority of clinical and economic papers referenced published over the past 10 years. However, the costs included in the model (and associated references for these costs) were outdated and we have now updated all costs included in the analysis (and references) to the most recent price year. Please see further response to this in response to the related comment from Reviewer 3. Thank you.

Response from authors to Reviewer 2:

(1) We have now completed Reference #9. Thank you also for your detailed comments provided in the attachment. We have now addressed the one suggestion for improvements made in this document. In the discussion section, we have expanded to clarify what our study has added to the evidence base (and differentiates our work from the wider ESTER report). We have said that the work presented in this paper has allowed us to focus on the economic viability of treatments specifically, as opposed to considering cost-effectiveness as part of a broader study into clinical effectiveness and safety. Additionally, and most notably, we have highlighted that the paper is an update on the previous cost-effectiveness results presented given that we have inflated our costs to a 2019 price year (previously 2016 price year) and so, this is an up-to-date assessment of the cost-effectiveness of alternative surgical treatments.

Response from authors to Reviewer 3:

(1) We have now updated our costs to a 2019 price year and have re-run all analyses to reflect the change in costs.

(2) Regarding the success rates for each of the re-treatments, we needed to make an assumption as to their success rate due to lack of appropriate data. We have now added text in the 'Model Structure' section to describe which surgical procedures would be administered as re-treatment, and we have also added text in the 'Effectiveness of surgical treatments' section to describe the assumption made around success of subsequent treatments. Thank you for highlighting this.

(3) Regarding presentation of absolute cure rates for all surgical interventions, our preference would be to leave the presentation of success rates in their current form. We have presented the absolute cure rates associated with RMUS at each of the different time points, and odds ratios of success associated with all comparators relative to RMUS. We feel that presenting a table showing the absolute rates associated with the eight comparator interventions would be overly cumbersome given that absolute rates of success are presented at multiple different time points and we would need to do the same for each one of the eight comparator surgeries. We feel that presenting the data in its current form is more efficient and should be easy to interpret by the reader.

(4) We have now given a rationale for this in the final sentence of the first paragraph in the Discussion.

(5) In the 'base-case analysis' section of the Results, we do explicitly say that over a lifetime time horizon RMUS is the least costly and second most effective intervention, behind Trad-Sling. However, we agree that this has not been emphasised enough and we have now added text in the discussion to explain that over 10-year and lifetime time horizons, Trad-sling also has a high probability of being cost-effective. We have now brought this out in the discussion section. As you mention, in the previous analysis Trad-sling appeared to be most cost-effective in the CEAC but having re-run the analysis, that is no longer the case and the CEAC has been updated to reflect this.

(6) This change has now been made (QALYs to three decimal places).

(7) Yes, we have now added these EVVPI results to an online supplementary file. Thank you.

(8) In the 'Complications and mortality rates after surgical treatments' section we describe that random effects meta-analysis models were fitted to data on complications after the different surgical procedures identified in the review of clinical effectiveness conducted as part of the wider study. We did not need to use Retro-MUS as the baseline intervention for this element as sufficient data were available from the identified studies. We have not gone into a huge amount of detail in this section on the methods and models used to derive the complication rates, however we feel that this would be providing unnecessary detail given the focus of the paper. We have now added a little more text in this section which should hopefully clarify the points raised. We only included the most frequently occurring complications, but where data did not exist on that complication for a specific procedure, it was assumed that that complication would not occur. Thank you.

(9) We have now addressed the formatting issues in all of the tables presented in the supplementary files. Thank you.

VERSION 2 – REVIEW

REVIEWER	Edna Keeney University of Bristol, UK
REVIEW RETURNED	10-Feb-2020
GENERAL COMMENTS	I am happy with the responses and edits made following my previous comments. Thank you.